# Production of Polyhydroxyalkanoates by Two Halophilic Archaeal Isolates from Chott El Jerid Using Inexpensive Carbon Sources

**DOI:** 10.3390/biom10010109

**Published:** 2020-01-08

**Authors:** Manel Ben Abdallah, Fatma Karray, Sami Sayadi

**Affiliations:** 1Laboratory of Environmental Bioprocesses, Centre of Biotechnology of Sfax, BP 1177, Sfax 3018, Tunisia; manelbenabdallah.cbs@gmail.com (M.B.A.); karray.fatma@gmail.com (F.K.); 2Center for Sustainable Development, College of Arts and Sciences, Qatar University, Doha 2713, Qatar

**Keywords:** polyhydroxyalkanoate, biodegradable polymer, hypersaline lake, PHA-producing archaea, carbon sources

## Abstract

The large use of conventional plastics has resulted in serious environmental problems. Polyhydroxyalkanoates represent a potent replacement to synthetic plastics because of their biodegradable nature. This study aimed to screen bacteria and archaea isolated from an extreme environment, the salt lake Chott El Jerid for the accumulation of these inclusions. Among them, two archaeal strains showed positive results with phenotypic and genotypic methods. Phylogenetic analysis, based on the 16S rRNA gene, indicated that polyhydroxyalkanoate (PHA)-producing archaeal isolates CEJGTEA101 and CEJEA36 were related to *Natrinema altunense* and *Haloterrigena jeotgali*, respectively. Gas chromatography and UV-visible spectrophotometric analyses revealed that the PHA were identified as polyhydroxybutyrate and polyhydroxyvalerate, respectively. According to gas chromatography analysis, the strain CEJGTEA101 produced maximum yield of 7 wt % at 37 °C; pH 6.5; 20% NaCl and the strain CEJEA36 produced 3.6 wt % at 37 °C; pH 7; 25% NaCl in a medium supplemented with 2% glucose. Under nutritionally optimal cultivation conditions, polymers were extracted from these strains and were determined by gravimetric analysis yielding PHA production of 35% and 25% of cell dry weight. In conclusion, optimization of PHA production from inexpensive industrial wastes and carbon sources has considerable interest for reducing costs and obtaining high yield.

## 1. Introduction

In recent decades, rapid population growth has increased the number of non-degradable materials. To solve this problem, biodegradable polymers such as polyhydroxyalkanoates (PHAs) have attracted immense interest for several applications in industry, medicine, and agriculture because of their main characteristics of biodegradability and biocompatibility that allow PHAs used as an alternative to petroleum-based plastics [1]. PHAs are produced by numerous microorganisms as insoluble storage inclusions in the cell cytoplasm, during unbalanced nutrient conditions [2]. Poly(3-hydroxybutyrate) (PHB) is the simplest homopolymer stored in prokaryotic cells [3]. This type is impermeable to oxygen, biodegradable, and biocompatible which make it a potential candidate for several applications [4]. Other PHA types, including poly(hydroxybutyrate-co-hydroxyvalerate) (PHBV) and polyhydroxyoctanoate (PHO) are evaluated mainly for biomedicine [5]. Although PHAs exhibit biodegradable and biocompatible properties that make them competitive in different applications, the high production cost of PHAs has become an obstacle. Therefore, the use of inexpensive carbon sources and waste products for PHA biosynthesis by halophiles is a good strategy and has been the subject of many investigations. 

Halophilic microorganisms are a potential source of PHAs. The production of PHA by bacteria has been extensively studied during the past decades [6]. However, scarce research has been devoted to study the diversity of aerobic PHA-producing archaea at high salinities. As of April 2016, the class Halobacteria contained 52 genera and 215 species. Most of them are aerobic extremely halophilic archaea, red-pink pigmented, neutrophilic, or alkaliphilic species. Currently, only a few haloarchaeal strains belonging to the genera *Halobacterium*, *Natronococcus*, *Natronobacterium*, *Halorubrum*, *Haloquadratum*, *Halococcus*, *Haloterrigena*, *Natrialba*, *Haloarcula*, and *Haloferax* were reported as producers of PHB and PHBV [7]. Halophilic archaea have received much interest because of their capacity to use a large variety of inexpensive and renewable carbon resources and so to reduce PHA production cost and obtain high productivity. Attempts have been made by examining the ability of halophilic archaeal strains to utilize inexpensive substrates. Synthesis of PHA by *Halogeometricum borinquense* strain TN9 or *Haloarcula* sp. IRU1, isolated from solar salterns or hypersaline lake using inexpensive substrates such as glucose or petrochemical wastewater were reported, respectively [8,9]. Halophilic archaea, employed as PHA producers, are also sources of hydrolytic enzymes, exopolysaccharides, and halocines [10]. 

In our previous article [11], we presented a report of hydrolytic enzymes production by 35 extremely halophilic strains isolated from a mixture of waters and sediments of the Tunisian hypersaline ephemeral lake, Chott El Jerid [11]. In order to further explore their potential biotechnological applications, the current work provided the first report exploring the PHA production by the same isolates described above from Chott El Jerid and evaluating the potential of promising strains to use inexpensive industrial wastes and carbohydrates. Since to our knowledge, one paper has addressed the production of PHA by archaeal strains isolated from Tunisian saltern [12], but never yet been studied in southern Tunisian chotts. The goal of this present investigation was to screen PHA-producing strains and evaluate the appropriate carbon sources for PHA production by optimizing growth conditions. It is also aimed to extract and quantify the polymer from promising strains. 

## 2. Materials and Methods

### 2.1. Sampling and Maintenance of Isolates

The sample S1-10, was collected from Chott El Jerid during the dry season as previously reported. The geographical position and the abiotic parameters (pH, salinity, temperature) were previously described [13]. Sixty-eight aerobic bacterial and archaeal isolates were previously obtained from the four enrichment cultures conditions. According to the enzymatic digestion profiles, 35 isolates were selected to test their capability of producing hydrolases. Taxonomic identification was achieved for 26 enzyme-producing isolates, through the 16S rRNA gene analysis [11].

### 2.2. Screening of Potential Halophilic PHA Producers by Staining Procedures

In the current research, with the aim of continuing the study of these previously isolated stains, we screened their biotechnological potential as PHA producers. The 35 bacterial and archaeal isolates described above were grown in PHA production medium containing (g L^−1^): 250 g NaCl, 10 g MgCl_2_. 6 H_2_O, 15 g MgSO_4_. 7 H_2_O, 4.0 g KCl, 1.0 g CaCl_2_. 2 H_2_O, 0.5 g NaHCO_3_, 1 g yeast extract, 10 g starch, 2% (*w*/*v*) agar at pH 6.8. All strains were grown at 37 °C on PHA detection agar containing 0.5 µg mL^−1^ Nile Red dye (Sigma-Aldrich). After 14 days of incubation, plates were exposed to UV light and the PHA accumulating colonies were detected [14]. Isolates were further tested by staining with Sudan Black B. The plates were kept undisturbed for 30 min and then rinsed gently by adding absolute ethanol. PHA producing colonies appeared black [15]. The bacterial type strain, *Escherichia coli*, as negative control was streaked onto medium plates incorporated with dye.

### 2.3. Screening of PHA Synthase Genes by Degenerate Polymerase Chain Reaction

Furthermore, the 35 isolates exhibited an amplification of PHA synthase genes. The detection of *PhaC1*/*C2* genes encoding for type II PHA synthase in bacteria domain was performed by amplification using forward and reverse primers: 179-L (5’- ACAGATCAACAAGTTCTACATCTTCGAC-3’) and 179-R (5’- GGTGTTGTCGTTGTTCCAGTAGAGGATGTC-3’), respectively [16]. The PCR program was as follows: 94 °C for 5 min, followed by 30 cycles of 94 °C for 1 min, 50 °C for 2 min, 72 °C for 2 min. The final step was at 72 °C for 5 min. In the domain Archaea, PHA synthases are composed of two subunits (*phaC*; *phaE*) and shared the closest identities with bacterial type III PHA synthases [17]. The detection of *phaC* and *phaE* genes was evaluated using Consensus Degenerate Hybrid Oligonucleotide Primers (CODEHOPs) for PCR amplification. The *PhaE* polymerase gene was amplified using the primers codehopEF (forward, 5’-CGACCGAGTTCCGCGAYATHTGGYT-3’) and codehopER (reverse, 5’-GCGTGCTGGCGGCKYTCNAVYTC-3’). The *PhaC* polymerase gene was amplified using the primers codehopCF (forward, 5’-ACCGACGTCGTCTACAAGGARAAYAARYT-3’) and codehopCR (reverse, 5’-GGTCGCGGACGACGTCNACRCARTT-3’) [17]. PCR was performed in a thermocycler (Applied Biosystems) in a 25 µL reaction volume containing 50 ng of target DNA, 1 x PCR Buffer, 0.2 µM of each primer, a 200 µM each DNTP, 1.25 U of Taq DNA polymerase (Fermentas). The following program for both units *PhaE* and *PhaC* consisted of one cycle of initial denaturation at 94 °C for 5 min; followed by 30 cycles of 94 °C for 30 s, 55 °C for 45 s, and 72 °C for 45 s; ending with 10 min at 72 °C. The PCR products (280 bp for *PhaC*; 230 bp for *PhaE*) were subjected to electrophoresis using 2% agarose gels. 

### 2.4. Staining of Promising Cells with Nile Red

According to screening tests using viable colony staining and molecular methods, two strains CEJGTEA101 and CEJEA36 were selected as PHA producers. Based on phylogenetic analysis, it has been previously reported that strains CEJGTEA101 and CEJEA36 exhibited 99% of sequence similarity with *Natrinema altunense* and *Haloterrigena jeotgali*, respectively [11]. Early stationary phase cells were prepared. Heat-fixed smears were washed with sterile distilled water, dried, then stained with 0.01% Nile Red in DMSO for 15–20 min. After staining, slides were washed with water and air-dried. The cells were viewed by an Olympus BX51 fluorescence microscope using "U-MWB2" mirror unit and observed with blue excitation wavelengths (460–490 nm) [18]. 

### 2.5. Optimization of Growth Conditions of the Potential Halophilic PHA Producers 

A 500 µL inoculum from a selected pre-culture was taken in exponential phase, then inoculated into 50 mL medium supplemented with starch, and cultivated at 37 °C with agitation rate of 180 rpm. Growth in PHA production medium was monitored spectrophotometrically at 600 nm each 24 h from 0 to 192 h. The effects of pH (5, 6, 6.5, 7, 8, and 9); temperature (20, 30, 37, 40, and 50 °C) and concentration of NaCl (0, 50, 100, 150, 200, 250, and 300 g L^−1^) on growth of selected isolates were carried out in PHA production medium as described above. The specific growth rates µ (h^−1^) were determined for the exponential phase of growth. 

### 2.6. Effect of Carbon Sources on PHA Production by Potential Halophilic PHA Producers 

The effect of starch, glucose, and industrial sugar waste on PHA production by selected strains were tested by using sugars (0%, 1%, 2%, 3%, 4% (*w*/*v*)) under optimized growth conditions. Carbon sources (industrial sugar waste and glucose) were filtered separately and supplemented to sterilized production medium in place of starch. The sugar waste used in this study was obtained from candy manufacturing industry in Sfax (Tunisia). The total organic carbon (TOC), total nitrogen (TN), total solids (TS), and volatile solids (VS) were determined according to standard methods [19]. The pH and electronic conductivity (EC) were measured using a NeoMet-type pH meter and a digital unit (Consort C831), respectively. Salinity was then determined by multiplying the EC by the correction factor (0.85). Cell density and PHA accumulation were achieved using these carbon sources.

### 2.7. Measurement of Cell Dry Weight (CDW) 

At optimized conditions, the cultures were centrifuged at 6000 rpm for 30 min. The cells were washed twice with sterile distilled water before lyophilization overnight to constant weight. 

### 2.8. Analysis of PHA in Dried Cells by Gas Chromatography (GC) 

About 4 mg lyophilized cells were mixed with 1 mL of chloroform, 0.85 mL of methanol, and 0.15 mL of sulfuric acid. The methanolysis was realized in a screw capped tube for 140 min at 100 °C. The resulting methylesters were analyzed with a Agilent Technologies 7890A chromatograph in a 30 m (5 % phenyl methyl siloxane) capillary column and a flame ionization detector. The program was used as previously reported [20]. Samples were analyzed in duplicate and calibration was performed using standard PHB (Sigma-Aldrich, USA). PHA content was defined as ratio of PHA weight to cell dry weight (terms of percentage). 

### 2.9. Polymer Extraction and Quantitative Analysis of PHA

PHA was recovered from dried cells using the sodium hypochlorite method according to Bhattacharyya et al., (2012) [21]. The pellet obtained was washed with distilled water, acetone and ethanol, respectively. The pellet was suspended in warm chloroform (60–65 °C) until evaporation. This represents the amount of PHA obtained after extraction from a known amount of dried cells (as percent PHA weight/CDW). 

After the chloroform had been evaporated, the polymer was converted to crotonic acid by heating in concentrated sulphuric acid for 10 min (crotonic acid assay). The polymer was quantified by spectrophotometry at 235 nm (Shimadzu UV spectrophotometer UV-1800) against a concentrated sulphuric acid as blank [22]. A standard curve was prepared using commercial Poly [®-3-hydroxybutyric acid] natural origin (Sigma-Aldrich) at concentrations ranging from 5–50 µg mL^−1^. 

## 3. Results

### 3.1. Screening of PHA-Producing Isolates from Chott El Jerid 

Thirty-five extremely halophilic isolates were previously obtained from hypersaline salt lake Chott El Jerid and were screened for enzyme production [11]. These previously isolated strains were screened for PHA accumulation by colony staining and molecular methods. Among them, two archaeal strains CEJGTEA101 and CEJEA36 showed presence of black granules when stained with the lipophilic dye Sudan Black B and bright orange fluorescence under UV transilluminator when stained with the specific dye Nile Red. Fluorescent colonies for the production of PHA were not detected for bacterial strains in our experiments (data of non PHA-accumulating strains were not shown here). Furthermore, the cells from CEJEA36 and CEJGTEA101 cultures showed brightly fluorescent orange granules as seen by fluorescence microscopy when stained with Nile Red, thus confirming the production of PHA.

These results were also confirmed by amplification of PHA synthase genes. Notably, only these two strains CEJGTEA101 (Figure 1a) and CEJEA36 (Figure 1b) showed bands with the predicted sizes of both *PhaE* (~230 bp) and *PhaC* (~280 bp). These two isolates were found to be PHA producers and were chosen for study in depth. 

Previous phylogenetic analysis revealed that these strains CEJGTEA101 and CEJEA36 were affiliated with *Natrinema altunense* and *Haloterrigena jeotgali*, respectively (99% similarity) [11]. 

### 3.2. Growth Kinetics of Potential Halophilic PHA Producers

The culture CEJGTEA101 was grown in PHA production medium as described above. After a lag phase of 1 day, the growth increased with a logarithmic phase of three days and finally reached the stationary phase on the fifth day and was maintained until the eighth day. Similarly, growth of the culture CEJEA36 in PHA production medium started with a lag phase of one day followed by a logarithmic phase of two days, and attained a short stationary phase on the fourth day after which a decline was rapidly observed till the eighth day.

### 3.3. Effect of Temperature, pH, Salinity on Growth of Potential Halophilic PHA Producers

The growth patterns of promising strains at different temperatures, pH and salinity in PHA production medium provided by starch were studied here. The effects of these parameters were studied to achieve higher PHA yield. The isolate grew at temperature range of 30–50 °C for strain CEJGTEA101 and 30–40 °C for strain CEJEA36. The optimum temperature for growth by both isolates was 37 °C. The growth increased till 37 °C and declined at temperatures extremes. The strain CEJGTEA101 supported NaCl for growth at concentrations ranging from 10%–30% (*w*/*v*) with an optimum growth at 20%. The strain CEJEA36 grew in medium containing 10%–30% (*w*/*v*) NaCl with an optimum at 25%. Both isolates didn’t grow at salinity under 100 g L^−1^. The pH range for growth of both strains CEJGTEA101 and CEJEA36 was 5–9 with an optimum 6.5 and 7, respectively (Figure 2a–c). This suggests that the maximum specific growth rates of the strain CEJEA36 were higher than that of the strain CEJGTEA101 in all parameters tested. 

### 3.4. Effect of Carbon Sources on Growth and PHA Production by Potential Halophilic PHA Producers

The chemical characteristics of sugar wastewater is given in Table 1.

In the present study, the promising isolates were chosen to study the production of PHA using a range of substrates (starch, glucose, and industrial sugar waste). The two strains behaved differently when grown on carbon sources. For the concentration of starch tested, 10 g L^−1^ and 20 g L^−1^ appeared the best concentrations providing the optimum growth for both strains CEJGTEA101 and CEJEA36, respectively. Both isolates grew with an optimum 20 g L^−1^ of glucose. The strain CEJGTEA101 found an optimum of growth at 30 g L^−1^ of industrial sugar waste, whereas, the strain CEJEA36 at 10 g L^−1^ (Figure 3d–f). 

Dry weights and percentage of PHA yields at stationary phases after 96, 120, and 144 h for the strain CEJGTEA101 and 72, 96, and 120 h for the strain CEJEA36 under optimal cultivation conditions are shown in Figure 4 and Figure 5, according to the gas chromatography analysis. *Natrinema altunense* strain CEJGTEA101 produced PHA when grown in all substrates tested. It was found that glucose was the best carbon source for PHA synthesis. This isolate showed the highest cell densities 380 ± 14 mg L^−1^ and the highest PHA content of 7% of CDW after 120 h of incubation. The PHA content was also higher (5.9% ± 0.23% of CDW) at 96 h and then decreased to reach 4.6% ± 0.2% of CDW after 144 h of incubation (Figure 4). Glucose is an easily assimilable carbon source that stimulates more production of PHA. The strain CEJGTEA101 could also accumulate PHA using starch with maximum level about 2.7% ± 0.14% of CDW after 120 h of incubation. It seems that industrial sugar waste, with 730 g L^−1^ of concentration, apparently could support the growth of the strain CEJGTEA101 (approxi 340–436 ± 36 mg L^−1^ of CDW) but did not contribute much for PHA production (approxi 0.38%–0.7% of CDW). 

The second isolate, *Haloterrigena jeotgali* strain CEJEA36, could produce PHA from the carbon substrates tested. The effect of carbon sources gave maximum PHA content of 2.36 ± 0.2%, 3.6%, and 2.91% of CDW when starch, glucose, and industrial waste were supplemented after 96 h of cultivation, respectively. The results showed a reduction in PHA production in the course of time after 120 h (Figure 5).

### 3.5. Yield of PHA Extracted from Potential Halophilic PHA Producers

After the incubation periods of 120 h and 96 h, the presence of polymers extracted from the two archaeal strains CEJGTEA101 and CEJEA36 grown on 2% of glucose (best substrate) was determined by gravimetric assay method yielding satisfactory amounts of PHA about 35% and 25% of its CDW, respectively (Table 2). It was observed that extracted PHA mass of strain CEJGTEA101 was higher in comparison with CEJEA36 (175 and 125 mg L^−1^, respectively). 

The productivity of PHA using glucose was 1.45 mg L^−1^ h^−1^ for strain CEJGTEA101 and 1.3 mg L^−1^ h^−1^ for strain CEJEA36. 

### 3.6. Polymer Characterization 

The nature of the PHA accumulated in cells of strains CEJGTEA101 and CEJEA36 was evaluated by gas chromatography analysis. According to Brandl et al., (1988), the identity of peaks was performed in comparison with polyester standards used under the same temperature gradient program as described above. The retention times of the peaks at 4.22 min and 5.82 min were identical to those of the 3-hydroxybutyrate and 3-hydroxyvalerate methyl esters, respectively. The GC spectrum of PHA product obtained by the isolate CEJGTEA101 showed a predominant peak appeared at retention time 4.6 minutes and indicated that this strain was a potential strain for producing poly(3-hydroxybutyrate) (PHB) of 7% of cell dry weight. The GC spectrum of PHA produced by the strain CEJEA36 showed a predominant peak at retention time at 5.7 minutes corresponding to poly(3-hydroxyvalerate) (PHV) of 3.6% of cell dry weight (Figure 6). The GC analysis indicated the ability of production SCL-PHA (short-chain-length) by these isolates. These results were further confirmed by the UV-visible spectrophotometric assay of polymer. The polymer obtained from the isolate CEJGTEA101 was hydrolyzed to crotonic acid which presented a characteristic peak at 235 nm similar to the standard PHB (Sigma-Aldrich) and therefore it can be proved that PHA produced by this archaeal isolate was a pure poly(3-hydroxybutyrate). The strain CEJEA36 presented a peak at 235 nm different to the standard PHB indicating another type of PHA (Appendix A).

## 4. Discussion

Several reports have focused on the isolation of large numbers of halophilic Archaea for use in various industrial processes. Thus, the discovery of novel compounds with great biotechnological potential may produce some interesting results [25]. Polyhydroxyalkanoates, biodegradable plastics, share similar properties with conventional synthetic plastics and consequently, PHAs could overcome solid waste management problems. In this regard, a range of polyhydroxyalkanoates are generated by these halophiles under unbalanced growth conditions using inexpensive renewable carbon sources as a strategy to reduce PHA production cost [26]. Researchers studied the usage of waste sources by halophilic archaea for producing PHA. Recently, a review highlighted that the most potent extremely halophilic archaeon, *Haloferax mediterranei* for PHA producer from inexpensive carbon-rich food and agro-industrial wastes [27]. As previously stated, the utilization of vinasse as a polluting waste generated from the ethanol industry leading to accumulation of 70% PHBV of CDW by *Haloferax mediterranei* [21] and 30% PHB of cell dry mass by *Haloarcula marismortui* [28].

In this study, two PHA producing archaea (CEJGTEA101 and CEJEA36) from Chott El Jerid, identified as *Natrinema altunense* and *Haloterrigena jeotgali*, accumulated PHB up to 35% and PHV up to 25% of CDW supplemented with 2% (*w*/*v*) glucose, respectively. Currently, no PHA was accumulated by the *Haloterrigena* species when glucose or starch was utilized as the sole carbon source. Only one haloarchaeon species, *Haloterrigena hispanica* strain FP1, was found to be able to accumulate intracellular PHB (0.13% of CDW) in highly saline medium containing 200 g L^−1^ when grown on carrot wastes as sole carbon source and used as an alternative for vegetable wastes management [7,24]. 

Similarly, a few of *Natrinema* spp. have potential to produce PHA using inexpensive carbon sources for improving biopolymers production. According to Danis et al., (2015), the species *Natrinema pallidum* strain 1KYS1 has been already reported to produce PHBV from cheap substrates such as melon, sucrose, apple, corn starch, whey, and tomato wastes. PHBV yield was reported at a level of 2.48%–53.14% of CDW (Table 2). In this current study, maximum CDW (436 ± 36 mg L^−1^) was shown when industrial sugar waste was utilized but not much for PHA accumulation (0.7 % of the dry cell weight) in the isolate CEJGTEA101 cells. Similarly, *Haloarcula japonica* strain T5 accumulated 1 wt % PHB of dried cells when growing on molasses, a by-product of sugar manufacturing industries [12]. Besides the use of waste products, glucose is currently being utilized by Haloarchaea for PHA accumulation except *Halobacterium*, *Natrialba,* and *Natronomonas* strains [29]. The isolate CEJGTEA101 could consume both starch and glucose as sole carbon source, however, utilization of glucose was more rapid than starch. Similar results of utilization of glucose by *Natrinema* spp., which accumulated PHB or PHBV per CDW up to 9.1% by *Natrinema altunense*, 22.9% by *Natrinema pallidum,* and 11.5% by *Natrinema pellirubrum* were observed [17]. Han and his colleagues showed a maximum yield in a shorter time (96 h of fermentation) compared to the time required by isolate CEJGTEA101. Additionally, it was notable that *Natrinema ajinwuensis* RM-G10 accumulated about 61% PHBV of its CDW [29]. These results showed that PHA content and its composition depend on species, growth conditions and type of substrate utilized. 

Although production of PHB or PHBV by members of *Natrinema* has been reported from pure sugars and waste materials, the detection of polyhydroxyalkanoates is a new observation in Tunisian hypersaline ecosystems, especially in southern salt lakes. What is notable in this study is that the unique isolate CEJGTEA101, belonging to the genus *Natrinema*, displayed parallel PHB and extracellular hydrolytic enzymes production when supplied with different carbon sources under high salinity [11]. This compares well with a unique report in which three strains isolated from Tunisian marine salterns, belonging to the genus *Haloarcula*, were observed to produce endopolymers (PHB) and exopolysaccharides. After growth on starch or glucose, strain T5 produced 0.5 wt % PHB of dried cells [12]. On the other hand, the extremely haloarchaeal isolate CEJEA36 did not show any hydrolytic activities, but, displayed PHV accumulating capacity and demonstrated more rapid growth than strain CEJGTEA101 in the same medium. The maximum growth and PHA content for both strains were observed at high salt concentration of 250 and 200 g L^−1^. However, the cultivation of the haloarchaeon *Natrinema ajinwuensis* RM-G10 in 300 g L^−1^ salt-containing medium demonstrated a decrease in growth and PHA accumulation [29] which is similar to our observation. 

## 5. Conclusions

PHB and PHV can be synthesized by two haloarchaeal species, belonging to the *Natrialbaceae* family which provide clear indications that PHA content and its composition depend on species, growth conditions and type of substrates utilized. Besides, this is the first report showing accumulation of PHV in *Haloterrigena* cells under high salinity. More importantly, the isolate CEJGTEA101 displayed parallel PHB and hydrolytic enzymes that may be a good candidate searching for other molecules. Future quantitative assays are required to improve the yield through the modification of growth conditions. Additionally, further research is needed to enlarge the possibility of polymer synthesis by other extremely halophilic archaea and bacteria using other substrates and industrial wastes in order to explore novel bioplastics from Chott El Jerid. 

## Figures and Tables

**Figure 1 biomolecules-10-00109-f001:**
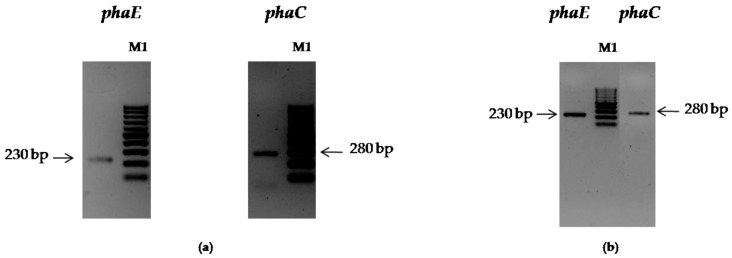
Amplification of the *phaE* and *phaC* genes of two strains CEJGTEA101 (**a**) and CEJEA36 (**b**). Lanes M1, 100-bp DNA marker.

**Figure 2 biomolecules-10-00109-f002:**
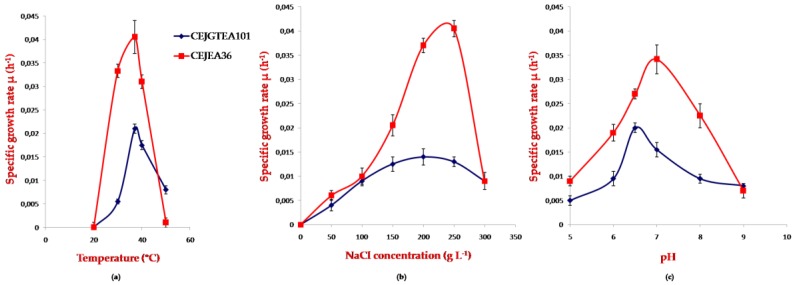
Effects of (**a**) temperature, (**b**) varying salinity, (**c**) pH on growth of strains CEJGTEA101 and CEJEA36 in polyhydroxyalkanoate (PHA)-accumulating medium. Data are shown as means of duplicate with their standard errors. µ: specific growth rate.

**Figure 3 biomolecules-10-00109-f003:**
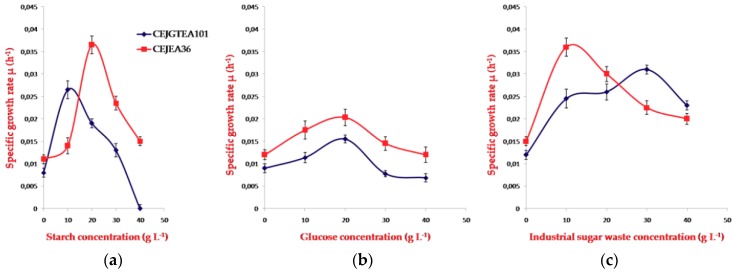
(**a**) Starch, (**b**) glucose, and (**c**) sugar waste concentration ranges for growth on PHA-accumulating medium of strains CEJGTEA101 and CEJEA36 under optimized conditions. Data are shown as means of duplicate with their standard errors. µ: specific growth rate.

**Figure 4 biomolecules-10-00109-f004:**
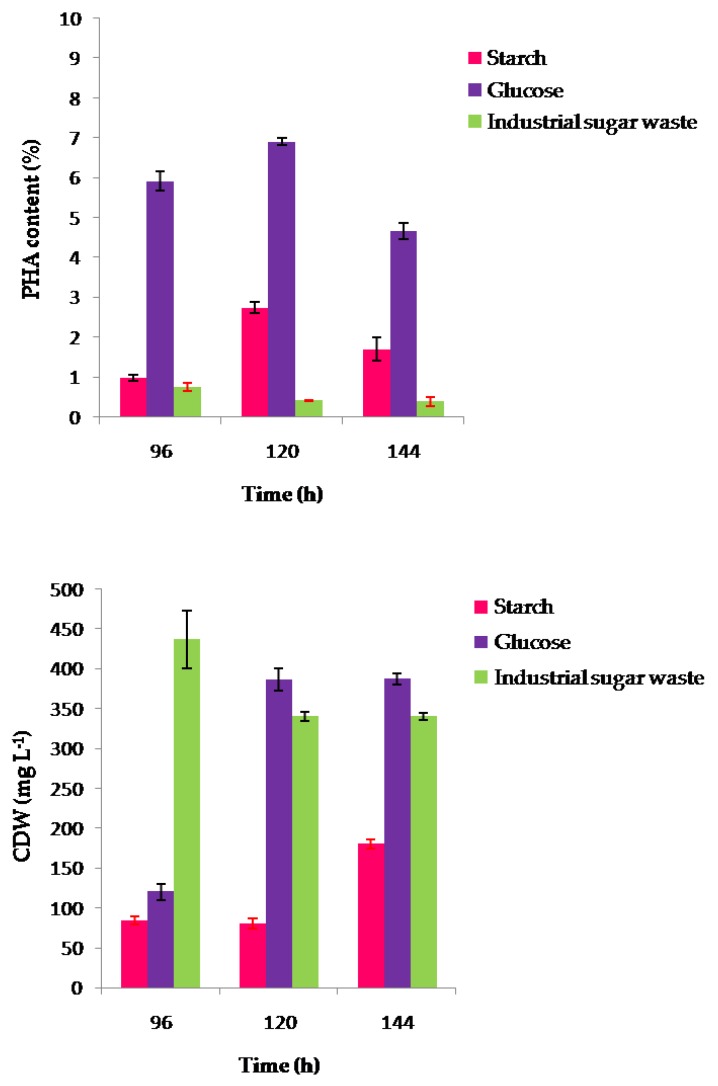
Effect of different carbon sources on CDW (mg L^−1^) and PHA content (wt %) of the strain CEJGTEA101 in a medium containing 10 g L^−1^ of starch or 20 g L^−1^ of glucose or 30 g L^−1^ of industrial sugar waste under optimal cultivation conditions: pH, 6.5; temperature, 37 °C; Concentration of sodium chloride, 20% upon 96 h, 120 h, and 144 h. CDW: cell dry weight; PHA: polyhydroxyalkanoate. Data are presented as means of duplicate with their standard errors. PHA content in dried cells was analyzed by gas chromatography.

**Figure 5 biomolecules-10-00109-f005:**
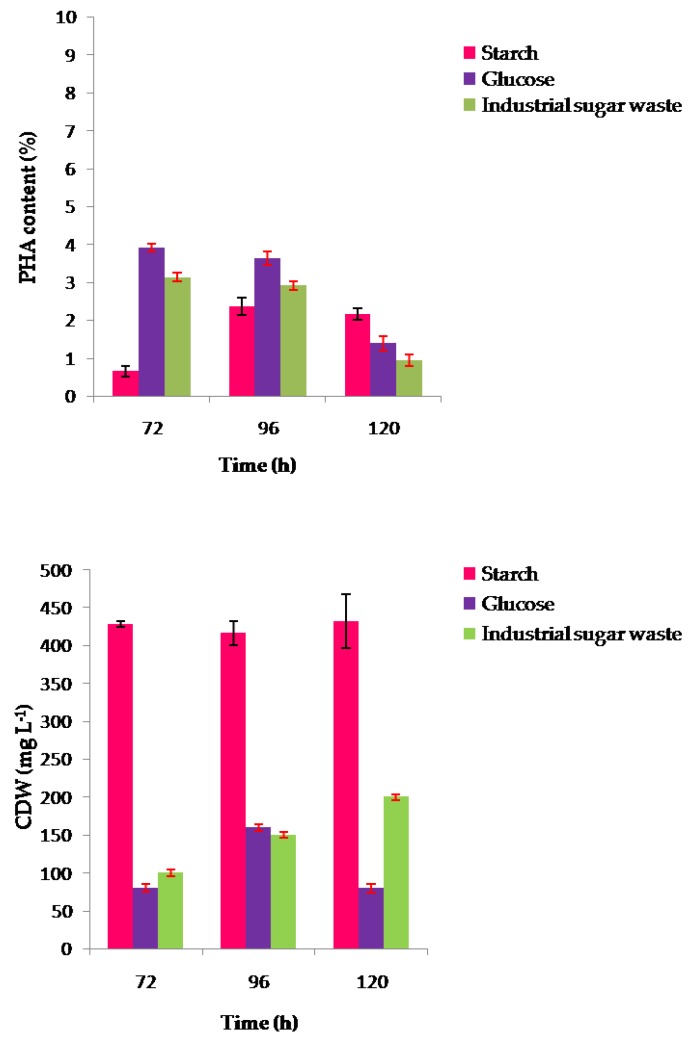
Effect of different carbon sources on CDW (mg L^−1^) and PHA content (wt %) of the strain CEJEA36 in a medium containing 20 g L^−1^ of starch or 20 g L^−1^ of glucose or 10 g L^−1^ of industrial sugar waste under optimal cultivation conditions: pH, 7; temperature, 37 °C; Concentration of sodium chloride, 25% upon 72 h, 96 h, and 120 h. CDW: cell dry weight, PHA: polyhydroxyalkanoate. Data are presented as means of duplicate with their standard errors. PHA content in dried cells was analyzed by gas chromatography.

**Figure 6 biomolecules-10-00109-f006:**
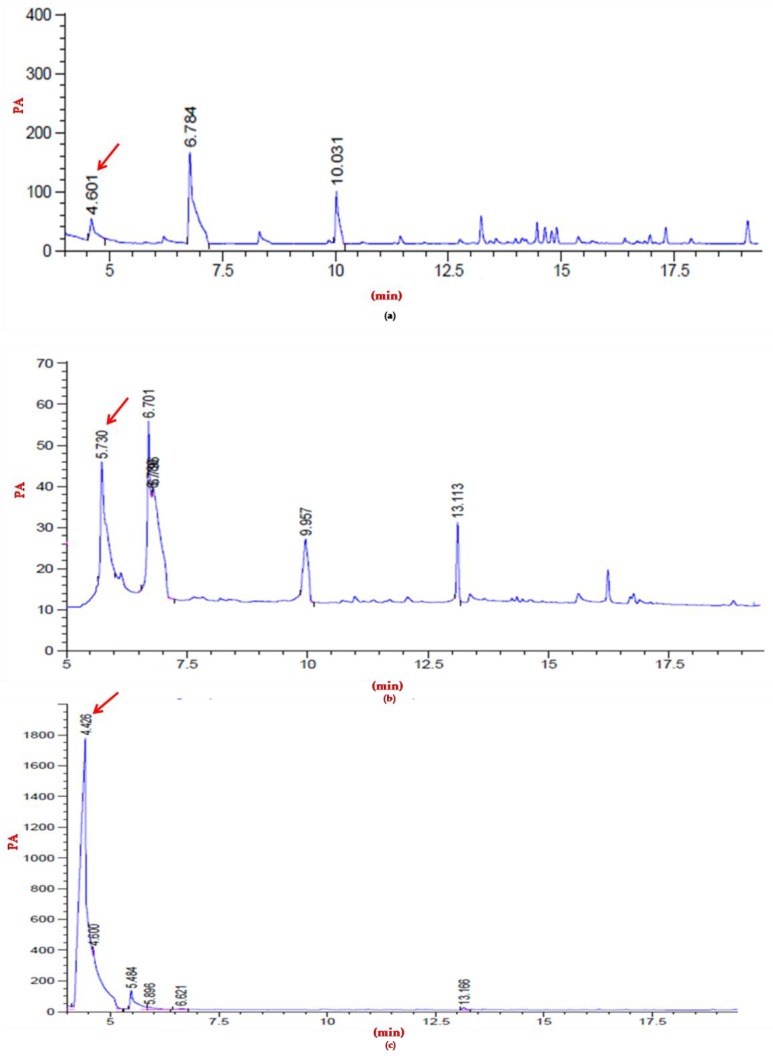
Gas chromatography (GC) analysis. Chromatograms of PHA obtained from cultures of the isolates (**a**) CEJGTEA101; (**b**) CEJEA36; and (**c**) PHB standard (Sigma). The peaks at 4.6 min and at 5.7 min represent the 3-hydroxybutyrate methylester and the 3-hydroxyvalerate methylester, respectively.

**Table 1 biomolecules-10-00109-t001:** Chemical composition of sugar wastewater.

Parameters	Sugar Wastewater
TOC (g L^−1^)	5.73 ± 0.8
TN (g L^−1^)	0.054 ± 0.02
TS (g L^−1^)	19 ± 1.2
VS (g L^−1^)	17.45 ± 0.8
pH	7.4
Salinity (g L^−1^)	22.06 ± 0.6

**Table 2 biomolecules-10-00109-t002:** Comparison of PHA production by two extremely haloarchaeal isolates CEJGTEA101 and CEJEA36 with closest phylogenetic relatives.

Isolates	Cell Dry Weight (g L^−1^)	Dry weight Extracted PHA ^a^ (g L^−1^)	PHA Accumulation (wt %)	Nature of PHA	Carbon Source	References
CEJGTEA101	0.5	0.175	35	PHB	Glucose	This study
*Natrinema pallidum* strain 1KYS1	0.174	0.075	53.14	PHBV	Corn starch	[23]
	2.219	0.055	2.48	PHBV	Sucrose	
	0.457	0.091	19.92	PHBV	Whey	
	0.371	0.039	10.5	PHBV	Melon	
	2.55	0.077	3.02	PHBV	Apple	
	3.858	0.464	12.03	PHBV	Tomato	
CEJEA36	0.5	0.125	25	PHV	Glucose	This study
*Haloterrigena hispanica* strain FP1	0.6	n.d.	0.13	PHB	Carrot waste	[24]

^a^ Values were determined by gravimetric method. The cells were cultured in PHA accumulation medium amended with 2% (*w*/*v*) glucose (best substrate) for 96 h (CEJEA36) and 120 h (CEJGTEA101) when maximum PHA production in these strains has been reached. PHA: polyhydroxyalkanoate, PHB: poly(3-hydroxybutyrate), PHV: poly(3-hydroxyvalerate), PHBV: poly(3-hydroxybutyrate-co-hydroxyvalerate), n.d.: not determined.

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
