# Peer review of "Production of Polyhydroxyalkanoates by Two Halophilic Archaeal Isolates from Chott El Jerid Using Inexpensive Carbon Sources"

_biomolecules, 2020, doi:10.3390/biom10010109_

Round 1

Reviewer 1 Report

     The authors should consider the following changes to the text of the manuscript:

1.) P.1,L.12:please change to "……...plastics has resulted in". Also, L.13: please insert "represent a" before "potent". Also, L.15:please delete "namely" and add a comma after "environment" (L.14). Also, L.24:please change to "gravimetric analysis yielding". Also, L.26: please substitute "inexpensive carbon sources" for "cheap" and delete "a" before "considerable" and delete "the" before "costs". Also, L.38: please insert "and" before "biocompatible". Also, L.40: please change "PHA" to "PHAs". Also, L.41: please change to "exhibit" and no punctuation after "properties".

2.) P.2,L.42: please change "sectors" to "applications". Also, please change to "cost of PHAs has become an obstacle". Also, please change "cheap" to "inexpensive". Also, L.46: please change  please change "have" to "has". Also, L.53: please change "cheap" to "inexpensive". Also, L.57: "cheap" to "inexpensive". Also, L.60: please add reference number after "article". Also, L.51: please add "the" before "Tunisian". Also, L.64: please change "potentiality" to "potential" and L.65: please change "cheap" to "inexpensive". Also, L.75: please write out "68". Also, L.81: please change to "we screened".

3.) P.3,L.91:please insert "the" before "PCR". Also, L.113: please delete "by" before "using" and delete "the" before "phylogenetic" in L.114. Also, L.127: please insert  "out" before "in".

4.) P.4,L.132: please change to "were tested by using sugars"...… Also, L.133: please change "carbone" to "carbon" and "filtrated" to "filtered". Also, L.134: please change to "in place of starch".Also, L.1354: just change to the name of the company followed by "Tunisia". Also, L.146:omit "the" before "standard". Also, L.149: please insert "the" before "sodium". Also, L.162: please write out "35" and L.170: please change "microscope" to "microscopy".

5.) P.5, figure 1: the authors should consider deleting figure 1 as it adds little to the manuscript. The use of these media can be summarized in the text.

6.) P.6, L.1197: Please change to "on the 5th day and was maintained until  the eighth day".

7.) P.7, Figure 4: These data should be deleted as they are too basic for this journal and can be summarized in the text. Figure 5A: the font is so small it is unreadable.

8.) P.8, Figure 5B. Again the font is unreadable. Also, L.236: delete "the" before "glucose".

9.) P.9, figure 6: why are there no standard error marks for nine of the bars?

10.) P.10, figure 7: many bars with no indication of standard errors.

11.) P.11, table 1: please change "Pollidum" to a lower case "p".

12.) P.12,L. 289: please insert "a" before "characteristic".

13.) P.13, figure 8: the lower two GC chromatograms need to be labeled as "b" and "c".

14.) P.14, L.299: please change to "Several reports have focused on the isolation of large numbers of halophilic Archaea for use in....". Also, L.300-301:  please change to "with great biotechnological potential may produce some interesting results." Also, please change "shared" to "share". Also, L.305: please change to "sources as a strategy to reduce PHA production cost (25). Also, L.313: delete "were". Also, L.322: lower case "p" for "Pollidum". Also, L.331: please substitute "more rapid" for "faster". Also, L.343: please insert "a" after "with" and L.344: please substitute "were observed" for "able" and L.345: please delete "this".

15.) P.15,L.347: please substitute "displayed" for "it showed" and "demonstrated more rapid growth than strain ……..".Also, L.349: please insert "were" before "observed". Also, L.350: please change to "medium demonstrated a decrease in growth and ……….". Also, L.353: please substitute "The" for "These".

16.) P.16,L.413: please change to "Environ."

Author Response

Dear Editor-in-chief,

I am Prof. Sami SAYADI, the corresponding author for the manuscript ID biomolecules-667659 entitled "Production of polyhydroxyalkanoates by two halophilic archaeal isolates from Chott El Jerid using inexpensive carbon sources"  which was submitted to Biomolecules.

We thank the reviewers for their careful reading of the manuscript and helpful comments and suggestions. We have revised our manuscript point by point according to their comments, as described below. 

Responses

Reviewer: 1

Comments and suggestions for Authors

1.) P.1,L.12:please change to "……...plastics has resulted in". Also, L.13: please insert "represent a" before "potent". Also, L.15:please delete "namely" and add a comma after "environment" (L.14). Also, L.24:please change to "gravimetric analysis yielding". Also, L.26: please substitute "inexpensive carbon sources" for "cheap" and delete "a" before "considerable" and delete "the" before "costs". Also, L.38: please insert "and" before "biocompatible". Also, L.40: please change "PHA" to "PHAs". Also, L.41: please change to "exhibit" and no punctuation after "properties".

Line 12: “.........plastics led to” was changed to “......plastics has resulted in”

Line 13: the words  “represent a” were inserted before ''potent'' and “are” was omitted.

Line 15: “namely” was deleted and a comma was added after environment (L.14).

Line 24: “gravimetric method yielding” was replaced by “gravimetric analysis yielding” 

Line 26: According to your comment and the comment of reviewer 3 in which he proposed to precise what kind of cheap substrate used,  “cheap” was replaced by “inexpensive industrial wastes and carbon”

              “a” before “considerable” was deleted

              “the” before “costs” was omitted

Line 38: “and” was inserted before “biocompatible”

Line 40: “PHA” was changed to “PHAs”

Line 41: “exhibited” was replaced by “exhibit”

                no punctuation after ''properties'' was done.

2.) P.2,L.42: please change "sectors" to "applications". Also, please change to "cost of PHAs has become an obstacle". Also, please change "cheap" to "inexpensive". Also, L.46: please change  please change "have" to "has". Also, L.53: please change "cheap" to "inexpensive". Also, L.57: "cheap" to "inexpensive". Also, L.60: please add reference number after "article". Also, L.51: please add "the" before "Tunisian". Also, L.64: please change "potentiality" to "potential" and L.65: please change "cheap" to "inexpensive". Also, L.75: please write out "68". Also, L.81: please change to "we screened".

Line 42: “sectors” was replaced by “applications”

              “cost of PHA became an obstacle”  was replaced by “cost of PHAs has    

               become an obstacle”

               “cheap ” was changed to “inexpensive”

Line 46: “have” was replaced to “has”

Line 53: “cheap ” was changed to “inexpensive”

Line 57: “cheap ” was changed to “inexpensive”

Line 60: The reference number [11] was added after “In our previous article”

Line 61: “the” was added before “Tunisian”

Line 64: “potentiality” was replaced by “potential”

Line 65: “cheap ” was changed to “inexpensive”

Line 75: “68” was written out as follows “Sixty-eight ”

Line 81: “we tried to screen” was changed to “we screened”

3.) P.3,L.91:please insert "the" before "PCR". Also, L.113: please delete "by" before "using" and delete "the" before "phylogenetic" in L.114. Also, L.127: please insert  "out" before "in".

Line 95: “The” was inserted before “PCR”

Line 113: “by” was deleted

Line 114: “The” was deleted

Line 127: “out” was inserted before “in” 

4.) P.4,L.132: please change to "were tested by using sugars"...… Also, L.133: please change "carbone" to "carbon" and "filtrated" to "filtered". Also, L.134: please change to "in place of starch".Also, L.1354: just change to the name of the company followed by "Tunisia". Also, L.146:omit "the" before "standard". Also, L.149: please insert "the" before "sodium". Also, L.162: please write out "35" and L.170: please change "microscope" to "microscopy".

Line 132: “was tested by incorporating separately of sugars” was modified as follows “were tested by using sugars

Line 133: “Carbone” was replaced by “Carbon”

                 “filtrated” was replaced by “filtered”

Line 134: “instead the starch” was replaced by “in place of starch”

Line 135: “food industry, sfax, Tunisa” was changed to the name of the company followed by Tunisia as follows  “candy manufacturing industry in Sfax (Tunisia)”

Line 146: “the” before “standard” was omitted

Line 149: “the” was inserted before “sodium” 

Line 162: “35” was written out as follows “Thirty-five”

Line 170: : “microscope” was changed to “microscopy”

5.) P.5, figure 1: the authors should consider deleting figure 1 as it adds little to the manuscript. The use of these media can be summarized in the text.

Figure 1 and the legend were deleted.

-In the text, “(Figure 1a,b)”; “(Figure 1c,d)”; “(Figure 1e,f)” were deleted (Lines 166, 167, 171)

6.) P.6, L.1197: Please change to "on the 5th day and was maintained until  the eighth day".

 Line 197: “on the 5th day lasted till the 8th day” was changed to “on the 5th day and was maintained until the eighth day”

7.) P.7, Figure 4: These data should be deleted as they are too basic for this journal and can be summarized in the text. Figure 5A: the font is so small it is unreadable.

Figure 4 and the legend were deleted.

The font of figure 5A was increased to be clear and readable  and the number of figure 5A was replaced Figure 2 (Line 217)

- In the text, “Figure 5A (a,b,c) “ was replaced by “Figure 2 (a,b,c) “ (Line 214)

8.) P.8, Figure 5B. Again the font is unreadable. Also, L.236: delete "the" before "glucose".

The font of figure 5Bwas increased to be clear and readable and the number of figure 5B was replaced “Figure 3“ (Line 229).

- In the text, “Figure 5B (d,e,f) “was replaced by “Figure 3 (d,e,f) “ (Line 227)

Line 236: “the” was deleted

9.) P.9, figure 6: why are there no standard error marks for nine of the bars?

There are standard error marks for all bars but some are so smaller that don't appear in the graph. that's why I increase the font of the figure and they appeared now all.

The number of figure 6 was replaced by “Figure 4“.

10.) P.10, figure 7: many bars with no indication of standard errors.

As the same, the font of the figure was increased and error bars were indicated.

The number of figure 7 was replaced by “figure 5“.

11.) P.11, table 1: please change "Pollidum" to a lower case "p".

In the table 1, the word “Pallidum” was replaced by “pallidum”

Table 1 was changed to Table 2.

12.) P.12,L. 289: please insert "a" before "characteristic".

Line 289: “a” was inserted before “characteristic”

13.) P.13, figure 8: the lower two GC chromatograms need to be labeled as "b" and "c".

The X-axis label “min” and (b) and (c) markings were added on the lower two GC chromatograms. The number of figure 8 was replaced by “figure 6“.

14.) P.14, L.299: please change to "Several reports have focused on the isolation of large numbers of halophilic Archaea for use in....". Also, L.300-301:  please change to "with great biotechnological potential may produce some interesting results." Also, please change "shared" to "share". Also, L.305: please change to "sources as a strategy to reduce PHA production cost (25). Also, L.313: delete "were". Also, L.322: lower case "p" for "Pollidum". Also, L.331: please substitute "more rapid" for "faster". Also, L.343: please insert "a" after "with" and L.344: please substitute "were observed" for "able" and L.345: please delete "this".

Line 299: the sentence “Several reports focused on isolation of large number of halophilic Archaea used in” was modified as follows: “Several reports have focused on the isolation of large numbers of halophilic Archaea for use in

Line 300-301: the sentence “with a great biotechnological constitutes an interesting research” was modified as follows “with great biotechnological potential may produce some interesting results 

“shared” was replaced by “share”

Line 305: the part of sentence “Sources as considered strategy to reduce the PHA production cost” was modified as follows: “sources as a strategy to reduce PHA production cost

Line 313: “were” was deleted

Line 322: the word “Pallidum” was replaced by “pallidum”

Line 331: “faster” was substituted by “more rapid”

Line 343: “a” was added after “with”

Line 344: “able” was replaced by “were observed”

Line 345: “this” was deleted.

15.) P.15,L.347: please substitute "displayed" for "it showed" and "demonstrated more rapid growth than strain ……..".Also, L.349: please insert "were" before "observed". Also, L.350: please change to "medium demonstrated a decrease in growth and ……….". Also, L.353: please substitute "The" for "These".

Line 347: “it showed” was replaced by “displayed”

“.It grew faster than the strain” was replaced by “and demonstrated more rapid growth than strain.....“

Line 349: “were” was inserted before “observed”

Line 350: The sentence “medium decreased the growth and” was modified as “medium demonstrated a decrease in growth and

Line 353: “these” was replaced by “The”

16.) P.16,L.413: please change to "Environ."

Line 413: “Envir” was changed to “Environ”

Reviewer 2 Report

The manuscript examined halophilic organisms isolated from an extreme environment (salt lake) for the accumulation of polyhydroxyalkanoates. The topic is important because polymers from the group of polyhydroxyalkanoates are a strong substitute for synthetic plastics, and what's more they are environmentally friendly and used in biomedicine. However, the high cost of producing PHAs is a major obstacle. Therefore, the authors of the publication proposed the use of halophilic organisms to produce PHAs and optimize these processes. The archaeal strains revealed great commercial and biotechnological values ​​with the aim of their use in fermentation processes using vegetable and agro-industrial wastes facilitating large-scale PHA production and cost reduction. However, as the authors of the publication emphasized, future quantitative assays are required to improve the yield through the modification of growth conditions, and further research is needed to increase the possibility of polymer synthesis by other halophilic organisms.
In my opinion, the manuscript can be published with minor revision:

1) Figure 8 needs improvement. The x-axis captions and (b) and (c) markings on the chromatograms are missing. In addition, current y and x axis captions are in too small font.

2) The signatures of Figures 3 and 8 need to be improved (double dot at the end).

3) Literature references have double numbering. This should be corrected.

Author Response

Dear Editor-in-chief,

I am Prof. Sami SAYADI, the corresponding author for the manuscript ID biomolecules-667659 entitled "Production of polyhydroxyalkanoates by two halophilic archaeal isolates from Chott El Jerid using inexpensive carbon sources"  which was submitted to Biomolecules.

We thank the reviewers for their careful reading of the manuscript and helpful comments and suggestions. We have revised our manuscript point by point according to their comments, as described below. 

Responses

Reviewer: 2

Comments and suggestions for Authors

1) Figure 8 needs improvement. The x-axis captions and (b) and (c) markings on the chromatograms are missing. In addition, current y and x axis captions are in too small font.

The X-axis label “min” and (b) and (c) markings were added on the lower two GC chromatograms. In addition, y and x axis captions are in big font and are readable. 

The number of figure 8 was replaced by “figure 6“.

2) The signatures of Figures 3 and 8 need to be improved (double dot at the end).

Figure 3 showed results of our previous published work, it doesn't add more to the manuscript. So, figure 3 was omitted.

In the figure 8, the second dot at the end of legend was deleted.

3) Literature references have double numbering. This should be corrected.

The numbering of literature references was corrected.

Reviewer 3 Report

The authors undertook an investigation concerning the production of polyhydroxyalkanotes (PHA) by two halophilic archaeal strains. Strains accumulating PHA were selected among 35 isolates obtained  from  Chott El Jerid site. Chromatographic analysis showed that strains of Natrinema altunense and Haloterrigena jeotgali were able to accumulate PHB and PHV using starch, glucose and sugar wastes. Yields of PHA were at the level of 35 and 25 % of cell dry weight, respectively.  Although the work is interesting and important in addressing PHA production by microorganisms, the quality of manuscript is not enough high to be published in Biomolecules journal. The main drawback of this manuscript is lack of novelty. It just another report about new strains having potential to synthesize PHA. To give imagination about production potential of these strains Authors should calculate productivity of PHA (eg. for 1 hour). Moreover, several critical issues have to be clarified to improve this manuscript.

Major comments

Abstract

Lines 19-20. The expression "spectrophotometric analyses revealed that the inclusions were identifies as ..." is wrong. Inclusions are cell structures. Authors identified PHA not inclusions as polyhydroxybutyrate and polyhydroxyvalerate.

Lines 21-25. It is not clear why Authors wrote that GC analysis showed that there is 7 and 3.6% of PHA in cells. And then they wrote that PHA yield is 35 and 25 %, respectively. The amount of extracted PHA should be lower than calculated on the base of GC.

Line 36. Authors should write in the abstract what kind of cheap substrate they used.

Line 65. Starch and glucose can not be considered as cheap substrates.

Line 131. Authors should describe and characterize the industrial sugar waste.

Line 154. The sentence "After the chloroform had been evaporated, the polymer granule was dissolved in.." is wrong. After extraction, PHAs are dissolved in chloroform - at this stage granules do not exit.

Figure 1. This figure should be described precisely otherwise it will be not comprehensible. Authors should point with the arrows where readers can see signals coming from PHA.

Figure 2. It is not clear why authors showed here the phylogenetic tree. If it is results of this work, the way in which tree was constructed should be described in the Materials and Methods chapter.

Lines 277-292. The proportion between polyhydroxybutyrate and polyhydroxyvalerate should be given.

Lines 352-359. Conclusions chapter should be modified. The first sentence can not be considered as conclusions of this work. This statement is obvious and it was known before writing this manuscript. I can agree with last sentence but it is not enough. Authors should write some conclusions on the base of results and eventually to give the direction of further studies.

Author Response

Dear Editor-in-chief,

I am Prof. Sami SAYADI, the corresponding author for the manuscript ID biomolecules-667659 entitled "Production of polyhydroxyalkanoates by two halophilic archaeal isolates from Chott El Jerid using inexpensive carbon sources"  which was submitted to Biomolecules.

We thank the reviewers for their careful reading of the manuscript and helpful comments and suggestions. We have revised our manuscript point by point according to their comments, as described below. 

Responses

Reviewer: 3

Comments and suggestions for Authors

The authors undertook an investigation concerning the production of polyhydroxyalkanotes (PHA) by two halophilic archaeal strains. Strains accumulating PHA were selected among 35 isolates obtained  from  Chott El Jerid site. Chromatographic analysis showed that strains of Natrinema altunense and Haloterrigena jeotgali were able to accumulate PHB and PHV using starch, glucose and sugar wastes. Yields of PHA were at the level of 35 and 25 % of cell dry weight, respectively.  Although the work is interesting and important in addressing PHA production by microorganisms, the quality of manuscript is not enough high to be published in Biomolecules journal. The main drawback of this manuscript is lack of novelty. It just another report about new strains having potential to synthesize PHA. To give imagination about production potential of these strains Authors should calculate productivity of PHA (eg. for 1 hour). Moreover, several critical issues have to be clarified to improve this manuscript.

Thank you for your comment and suggestion.

The authors tried to improve the quality of manuscript according to the reviewer's comments. Indeed, this work is the first report about new archaeal strains from Tunisian Sahara Desert,  having potential to synthesize polymers from the group of polyhydroxyalkanoates which are a strong substitute for synthetic plastics. Besides the originality of Chott El Jerid, It is interesting to note that the usage of industrial wastes/carbohydrates by halophilic archaea facilitating PHA production and cost reduction have never yet been studied in southern Tunisian chotts.

Major comments

The abstract

Lines 19-20. The expression "spectrophotometric analyses revealed that the inclusions were identifies as ..." is wrong. Inclusions are cell structures. Authors identified PHA not inclusions as polyhydroxybutyrate and polyhydroxyvalerate.

Line 19-20:  the expression “spectrophotometric analyses revealed that the inclusions were identifies as ...” was replaced by “spectrophotometric analyses revealed that the PHA were identified as polyhydroxybutyrate and polyhydroxyvalerate

Lines 21-25. It is not clear why Authors wrote that GC analysis showed that there is 7 and 3.6% of PHA in cells. And then they wrote that PHA yield is 35 and 25 %, respectively. The amount of extracted PHA should be lower than calculated on the base of GC.

In this work, two methods used in the quantitative analysis of PHA. Firstly, the authors used GC analysis ( methanolysis method) from lyophilized cells to optimize the best condition of growth cultivation and PHA production. Under nutritionally optimal cultivation conditions, the dry cells of promising isolates were subjected to second method (gravimetric method) using extraction with chloroform and then we found 35 % and 25 % of Cell dry weight.

Line 36. Authors should write in the abstract what kind of cheap substrate they used.

Line 26: The kind of inexpensive substrates used are sugar wastewater and carbohydrates.

The words “cheap sources” were replaced by the expression “inexpensive industrial wastes and carbon sources

Introduction

Line 65. Starch and glucose cannot be considered as cheap substrates.

Line 65: the expression “cheap substrates such as starch, glucose and industrial sugar wastes” was modified as follows: “inexpensive industrial wastes and carbohydrates

Materials and Methods

Line 131. Authors should describe and characterize the industrial sugar waste.

Thank you for your comment. We agree with you in this point.

In Materials and Methods section, We presented the description and analytic methods in this concern with some corrections given by the reviewer 1.

Line 131-135:  The paragraph “The effect of starch, glucose and industrial sugar waste on PHA production by selected strains was tested by incorporating separately of sugars (0, 1, 2, 3, 4% (w/v)) under optimized growth conditions. Carbone sources (industrial sugar waste and glucose) were filtrated separately and supplemented to sterilized production medium instead the starch. The sugar waste used in this study was obtained from food industry, Sfax, Tunisia and the pH was 7.4.” was modified as follows:

The effect of starch, glucose and industrial sugar waste on PHA production by selected strains were tested by using sugars (0, 1, 2, 3, 4% (w/v)) under optimized growth conditions. Carbon sources (industrial sugar waste and glucose) were filtered separately and supplemented to sterilized production medium in place of starch. The sugar waste used in this study was obtained from candy manufacturing industry in Sfax (Tunisia). The Total Organic Carbon (TOC), Total Nitrogen (TN), Total Solids (TS), Volatile Solids (VS) were determined according to standard methods (reference). The pH and Electronic conductivity (EC) were measured using a NeoMet-type pH meter and a digital unit (Consort C831), respectively. Salinity was then determined by multiplying the EC by the correction factor (0.85).

Line 135: The sentence “and the pH was 7.4” was removed from this paragraph and placed in the Table 1

In Results section, We presented the chemical composition of sugar wastewater in Table 1.

 Line 221: We added  before “In the present study” this sentence and title of Table 1 and Table 1 as follows:

“The chemical characteristics of sugar wastewater is given in Table 1. ”

Table 1. Chemical composition of sugar wastewater

Parameters

Sugar wastewater

TOC  (g L-1)

5.73 ± 0.8

TN     (g L-1)

0.054 ± 0.02

TS      (g L-1)

19 ± 1.2

VS      (g L-1)

17.45 ± 0.8

pH

7.4

Salinity (g L-1)

22.06 ± 0.6

Line 154. The sentence "After the chloroform had been evaporated, the polymer granule was dissolved in.." is wrong. After extraction, PHAs are dissolved in chloroform - at this stage granules do not exit.

Line 154-156: In Materials and Methods section, the sentence “After the chloroform had been evaporated, the polymer granule was dissolved in concentrated 154 sulphuric acid and incubated at 100 °C in boiling water bath for 10 min to convert PHA to crotonic 155 acid (crotonic acid assay)” was modified as follows: “After the chloroform had been evaporated, the polymer was converted to crotonic acid by heating in concentrated sulphuric acid for 10 min (crotonic acid assay).

Results

Figure 1. This figure should be described precisely otherwise it will be not comprehensible. Authors should point with the arrows where readers can see signals coming from PHA.

Thank you for your comment. Figure 1 and the legend were deleted as it adds little to the manuscript in agreement with the comment of Reviewer 1. 

Figure 2. It is not clear why authors showed here the phylogenetic tree. If it is results of this work, the way in which tree was constructed should be described in the Materials and Methods chapter.

Thank you for your comment. The phylogenetic tree presented results of previous work, so, figure 3 was omitted.

Lines 277-292. The proportion between polyhydroxybutyrate and polyhydroxyvalerate should be given

Line 284: the expression “of 7 % of cell dry weight“ was added

Line 286: the expression “of 3.6 % of cell dry weight“ was added

Conclusion

Lines 352-359. Conclusions chapter should be modified. The first sentence can not be considered as conclusions of this work. This statement is obvious and it was known before writing this manuscript. I can agree with last sentence but it is not enough. Authors should write some conclusions on the base of results and eventually to give the direction of further studies.

Thank you for your comment.

Line  353-355 : The first sentence of conclusion “These archaeal strains obtained from this study revealed great commercial and  biotechnological values with the aim of their use in fermentation processes using vegetable and agro-industrial wastes facilitating large-scale PHA production and cost reduction.” was modified as follows  “PHB and PHV can be synthesized by two haloarchaeal species, belonging to the Natrialbaceae family which provide clear indications that PHA content and its composition depend on species, growth conditions and type of substrates utilized. Besides, this is the first report showing accumulation of PHV in Haloterrigena cells under high salinity. More importantly, the isolate CEJGTEA101 displayed parallel PHB and hydrolytic enzymes that may be a good candidate searching for other molecules.

Other modifications:

-In the text, “Table 1” was replaced by “Table 2” (Lines 268, 270).

-In the text, “Figure 2a and Figure 2b” were replaced by “Figure 1a and Figure 1b, respectively” (Line 180)

- “Figure 2“ was replaced by “Figure 1“ (Line 184).

-“Figure 6“ was replaced by “Figure 4“ (Line 239, 246)

-“Figure 7“ was replaced by “Figure 5“ (Line 256, 258)

-“Figure 8“ was replaced by “Figure 6“ (Line 286, 294)

-Line 12: “uses“ was replaced by “use“

-Line 366: “NB., HZ.,“ were deleted.

- Determination of productivity of PHA (according to suggestion of reviewer 3)

Line 276: We calculate the productivity, and the sentence “The productivity of PHA using glucose was 1.45 mg L-1 h-1 for strain CEJGTEA101 and 1.3 mg L-1 h-1 for strain CEJEA36. “ was added.

References

-This reference was added in the references list:

APHA.; AWWA.; WEF. Standard methods for the examination of water and wastewater, 23 nd ed.;  American Public Health Association, American Water Works Association, Water Environment Federation, Washington DC, 2005; pp. 1-1504.

Line 439: These words “Use”; “Whey”; “Polyhydroxyalkanoates” and “Production” were changed to a lower case “use”; “whey”; “polyhydroxyalkanoates” and “production” 

Line 441: These words “Biosynthesis''; “Edge”; “Water” ; “Activity-Haloarchaea”;  “Biopolyester” and “Factories” were changed to a lower case “biosynthesis''; “edge”; “water” ; “activity-haloarchaea”;  “biopolyester” and “factories”

Round 2

Reviewer 3 Report

The manuscript was significantly improved by Authors and could be considered for publication in Biomolecules. One thing that can be done to improve the quality of manuscript is enhancing the novelty description in the Introduction chapter.